# EXTENDING MULTI-MODAL CONTRASTIVE REPRESENTATIONS

## ABSTRACT

Multi-modal contrastive representation (MCR) of more than three modalities is critical in multi-modal learning. Although recent methods showcase impressive achievements, the high dependence on large-scale, high-quality paired data and the expensive training costs limit their further development. Inspired by recent C-MCR, this paper proposes **Ex**tending **M**ultimodal **C**ontrastive **R**epresentation (Ex-MCR), a training-efficient and paired-data-free method to flexibly learn unified contrastive representation space for more than three modalities by integrating the knowledge of existing spaces. Specifically, Ex-MCR aligns multiple existing MCRs into the same base-MCR, which can effectively preserve the original semantic alignment of the base-MCR. Besides, we comprehensively enhance the entire alignment learning pipeline from the perspectives of training data, architecture, and learning objectives. With the preserved original modality alignment and the enhanced space alignment, Ex-MCR shows superior representation learning performance and excellent modality extensibility. To demonstrate the effectiveness of Ex-MCR, we extend pre-trained audio-text and 3D-image representations into existing vision-text spaces, leveraging the overlapping text and image modality, respectively. Remarkably, without using any paired data, Ex-MCR learns a 3D-image-text-audio unified contrastive representation, and it achieves state-of-the-art performance on audio-visual, 3D-image, audio-text, visual-text retrieval, and 3D object classification tasks. More importantly, extensive qualitative results further demonstrate the emergent semantic alignment between the extended modalities (e.g., audio and 3D), which highlights the great potential of modality extensibility.

## 1 INTRODUCTION

Multi-modal Contrastive Representation (MCR) learning endeavors to align inputs from diverse modalities within a shared representation space. Recently, the high-quality contrastive representations of more than three modalities attract increasing attention (Girdhar et al., 2023; Guzhov et al., 2022; Xue et al., 2023a;b; Liu et al., 2023b; Hegde et al., 2023; Guo et al., 2023), and play a fundamental role in many application scenarios of multi-modal understanding (Su et al., 2023; Zhang et al., 2023; Zhao et al., 2023; Wang et al., 2023a; Han et al., 2023) and generation (Tang et al., 2023; Liu et al., 2023a; Ramesh et al., 2022; Rombach et al., 2022; Gafni et al., 2022; Huang et al., 2023a). Despite the achievements of multi-modal contrastive learning, its broader and more flexible application is still constrained by the high dependence on large-scale, high-quality paired data and extremely costly training resources.

Recently, Wang et al. (2023b) introduces a novel training-efficient method, called C-MCR, for learning contrastive representations between modalities that lack paired data by mining knowledge from existing MCR spaces. It connects two pre-trained spaces onto a new shared space via overlapping modalities. Since the modalities of pre-trained spaces are intrinsically aligned, the connection learned from overlapping modalities can also be transferred to non-overlapping modalities. Experimentally, without using image-audio and 3D-text data pairs, C-MCR demonstrates advanced performance in image-audio and 3D-text downstream tasks.

Despite the remarkable flexibility and performance of C-MCR, its broader applications are hindered by a critical limitation: C-MCR mainly focuses on learning a new space for the two non-overlapping

modalities, while the modality alignments in powerful original pre-trained spaces are forgotten. As a result, C-MCR faces challenges in concurrently establishing connections among three or more MCRs. Therefore, it can not be used to flexibly learn unified contrastive representation space for more than three modalities.

This paper introduces **Ex**tending **M**ulti-modal **C**ontrastive **R**epresentations (Ex-MCR), a novel training-efficient and paired-data-free unified representation learning method with excellent modality extensibility. Ex-MCR better preserves the alignment within the original pre-trained space and enhances the overall learning pipeline to align different spaces more robustly. Specifically, the two important designs of Ex-MCR are discussed in detail below:

Firstly, we extend one MCR space (called leaf-MCR) into another fixed MCR space (called base-MCR) rather than connecting two pre-trained spaces to a new space. Such a simple yet effective approach maximizes the preservation of modality alignment within base-MCR, demonstrating great potential for integrating more pre-trained spaces.

Secondly, we enhance the whole learning process to promote stronger alignment across different spaces. Specifically: 1) From the training data perspective, we extract various modality-centric pseudo data. Since one modality can not be fully represented by another modality (such as audio embeddings retrieved by images can not reflect the whole semantic space of audio), we combine the pseudo pairs retrieved by different modalities to prompt more generalizable alignments. 2) From the architecture perspective, we propose a decoupled projector, which reduces interference among different optimization objectives. 3) From the learning objective perspective, we employ a dense contrastive loss on pseudo-pairs between all possible modalities pairs, further enhancing the stability of learned alignments.

Utilizing Ex-MCR, we can flexibly align multiple leaf-MCR spaces onto a same base-MCR space without any paired data and with extremely low training costs. To evaluate the effectiveness of our Ex-MCR, we try to extend pre-trained 3D-image and audio-text spaces onto image-text space via the overlapping image and text modality, which derive unified audio-image-text-3D representations. Without using any paired data, Ex-MCR attains state-of-the-art performance results across various zero-shot tasks, including audio-visual, 3D-image, audio-text, visual-text retrieval, and 3D object classification. More importantly, semantic alignment is also observed between extended modalities (e.g., audio-3D), which highlights the potential of Ex-MCR in modality extensibility.

Our contributions can be summarized as three-fold:

(1) We propose **Ex**tending **M**ulti-modal **C**ontrastive **R**epresentations (Ex-MCR), a novel training-efficient and paired-data-free representation learning method for more than three modalities.

(2) We comprehensively enhance the entire space alignment learning pipeline from the perspectives of training data, architecture, and learning objectives. These novel designs offer valuable insights about effectively integrating knowledge within existing MCRs.

(3) We obtain high-quality unified audio-image-text-3D representations using Ex-MCR, which exhibits advanced performance on a series of tasks and excellent modality scalability. Besides, we also conduct detailed ablation studies to verify the effectiveness of each proposed component.

## 2 RELATED WORKS

### 2.1 MULTI-MODAL CONTRASTIVE REPRESENTATIONS

Multi-modal Contrastive Representations (MCR) learning aims to acquire semantically aligned cross-modal representations by pretraining the model on large-scale paired data. These aligned representations play a pivotal role in downstream comprehension and generation tasks. Inspired by the success of CLIP (Radford et al., 2021), many works try to learning contrative representations for two modalities (Radford et al., 2021; Li et al., 2022; 2021; Gan et al., 2022; Xu et al., 2021). CLIP (Radford et al., 2021) and ALIGN (Jia et al., 2021) learn shared vision-text representations from million-level image-text pairs. CLAP (Elizalde et al., 2023; Wu et al., 2023) learns the audio-text representation, and CAV-MAE (Gong et al., 2022) focus on acquiring shared audio-visual feature space. C-MCR (Wang et al., 2023b) focuses on learning new representation space by connecting the pre-trained spaces through overlapping modality.

Apart from aligning two modalities, shared representations for more than three modalities attract increasing attention. AudioCLIP (Guzhov et al., 2022) and WAV2CLIP (Wu et al., 2022) train an audio encoder aligned with CLIP using audio-text-image triplets data. ULIP (Xue et al., 2023a;b) and openshape (Liu et al., 2023b) construct 3D-image-text triplets data through rendering 3D mesh into 2D images and captioning images for textual description, thereby learning a corresponding 3D encoder for image-text MCR space. Furthermore, Imagebind (Han et al., 2023) exclusively utilizes data pairs between various modalities and images to expand CLIP with multiple modal alignment encoders.

However, these methods heavily rely on large-scale, high-quality paired data collected from the internet or generated automatically and exceptionally high computational resources. Due to the lack of high-quality paired data for more modal combinations, such as audio-visual and text-3D, the extensibility of representation learning is notably constrained. Furthermore, the exceedingly high computational costs also diminish the flexibility of MCR learning.

## 2.2 AUDIO-VISUAL AND 3D-TEXT LEARNING

Audio-vision and 3D-text learning have significant applications in multi-modal recognition (Gemmeke et al., 2017; Chen et al., 2020b; Chang et al., 2015; Dai et al., 2017), localization (Chen et al., 2020a; Achlioptas et al., 2020; Zhao et al., 2021; 2018; Mo & Morgado, 2022; Chen et al., 2021), question-answer (Wang et al., 2023a; Zhao et al., 2023; Azuma et al., 2022; Lin et al., 2023b), and generation (Ruan et al., 2023; Poole et al., 2022; Lin et al., 2023a). They also play important roles in robot-related tasks such as human-machine interaction and synthetical information obtaining in complex environments (Peng et al., 2023; Huang et al., 2023b).

However, audio-visual datasets (Gemmeke et al., 2017; Chen et al., 2020b) often suffer from substantial noise due to soundless objects and invisible sounds. Additionally, paired 3D-text data (Chang et al., 2015) is scarce and expensive to collect. The scarcity of large-scale datasets hampers the further advancement of 3D-text and audio-vision contrastive representations. Previous methods, such as AudioCLIP (Guzhov et al., 2022) and ULIP (Xue et al., 2023a;b), mainly focus on automatically collecting or generating more paired data, but they are still limited by the relatively low quality of the training datasets. Our approach overcomes the reliance on paired data, achieving superior performance in audio-vision and 3D-text retrieval without using any audio-vision or 3D-text data.

## 3 EXTENDING MULTI-MODAL CONTRASTIVE REPRESENTATIONS

### 3.1 EXTENDING RATHER THAN CONNECTING

Given two pre-trained MCR spaces on modalities $(\mathcal{A}, \mathcal{B})$ and $(\mathcal{B}, \mathcal{C})$, C-MCR (Wang et al., 2023b) employs two projectors to map them into a new shared space, where the alignment of different spaces can be learned from overlapping modality $\mathcal{B}$. Since each pre-trained space intrinsically contains the alignment of $(\mathcal{A}, \mathcal{B})$ and $(\mathcal{B}, \mathcal{C})$, the alignment learned from overlapping modality theoretically can be transferred to the non-overlapping modalities.

Specifically, for aligning different spaces, the embeddings of $\mathcal{B}$ are aligned in the new space, and pseudo $(\mathcal{A}, \mathcal{C})$ pairs retrieved by the same data of $\mathcal{B}$ are also aligned for a more comprehensive inter-space alignment. Moreover, the embeddings of different modalities within a same space are realigned to close the modality gap (Liang et al., 2022), which significantly enhances the transferability of learned inter-space alignment. C-MCR shows remarkable flexibility and versatility since connecting two existing spaces only requires two learnable projectors and unpaired unimodal data.

However, C-MCR mainly focuses on learning a new space for the two non-overlapping modalities $(\mathcal{A}, \mathcal{C})$, while the original modality alignment in pre-trained spaces is forgotten. As a result, it faces challenges in concurrently establishing connections among three or more spaces. Therefore, C-MCR is not suitable for learning a unified representation space for more than three modalities.

To learn unified multi-modal representations in a training-efficient and paired-data-free manner, we propose to extend one space into another space rather than connect two spaces to a new space. Considering the two spaces on modalities $(\mathcal{A}, \mathcal{B})$ and $(\mathcal{B}, \mathcal{C})$, Ex-MCR chooses one as the base-

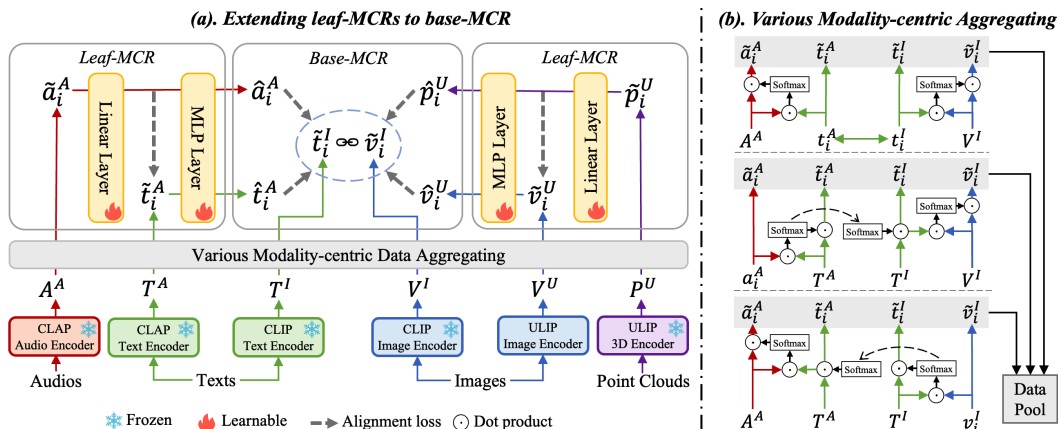

Figure 1: **The pipeline of Ex-MCR.** (a) We extend leaf-MCRs to base-MCR via the overlapping modalities. The base-MCR is frozen and the leaf-MCRs are aligned to base-MCR via projectors. (b) When extending the audio-text space to the text-image space, we iteratively use texts, audios, and images as queries to retrieve and aggregate the corresponding semantically consistent embeddings. The pseudo embedding pairs generated from different modality data are shuffled together to build the final various modality-centric data pool.

MCR $(\mathcal{A}, \mathcal{B})$, and the other as the leaf-MCR $(\mathcal{B}, \mathcal{C})$. In the "Extending" scheme, the base-MCR space is frozen, and we only train one projector to map leaf-MCR to base-MCR via the overlapping modalities $\mathcal{B}$. Specifically, we employ the native pairs of $\mathcal{B}$ and pseudo pairs generated by $\mathcal{B}$ to align leaf-MCR to base-MCR. Simultaneously, we close the modality gap between $(\mathcal{B}, \mathcal{C})$ modalities of leaf-MCR, thereby facilitating more transferable alignments.

In contrast to C-MCR, Ex-MCR can conveniently expand more spaces and learn unified representation for three or more modalities. Benefiting from efficient training and no need of paired data, we can flexibly align multiple leaf-MCR spaces to a same base-MCR space. In addition to explicitly establishing alignment among modalities of leaf-MCR and base-MCR, semantic alignment also emerges between extended modalities. Ex-MCR employs base-MCR as a bridge for achieving semantic alignment among modalities in multiple leaf-MCR spaces.

### 3.2 ENHANCING ALIGNMENT LEARNING PIPELINE

Before delving into the details of our learning pipeline, we first clarify the necessary symbols and notations. We align the ULIP (3D-image) and CLAP (audio-text) onto CLIP (image-text). As shown in Fig. 1 (a), the unimodal data of audios $A$, texts $T$, images $V$, and 3D point clouds $P$ are input to their corresponding encoders, and the set of extracted feature are denoted as $\mathbf{A}^A$, $\mathbf{T}^A$, $\mathbf{T}^I$, $\mathbf{V}^I$, $\mathbf{V}^U$ and $\mathbf{P}^U$, where superscripts $^A$, $^I$, $^U$ indicate representation space of CLAP, CLIP, ULIP, respectively. The $\mathbf{A}^A = \{\mathbf{a}_1^A, \mathbf{a}_2^A, \dots \mathbf{a}_{n_a}^A\}$ where $n_a$ is the number of all audio data and $\mathbf{a}_i^A$ represents the CLAP feature of $i$-th audio. Similarly, there are $\mathbf{t}_i^A, \mathbf{t}_i^I, \mathbf{v}_i^I, \mathbf{v}_i^U, \mathbf{o}_i^U$ in $\mathbf{T}^A, \mathbf{T}^I, \mathbf{V}^I, \mathbf{V}^U, \mathbf{P}^U$ respectively.

In Ex-MCR, freezing base-MCR allows us to maintain the original alignment of base-MCR but also implies that the modality gap within base-MCR is preserved. Consequently, it becomes necessary to map the leaf-MCR to more suitable positions within base-MCR space. To this end, we enhance the entire alignment learning pipeline from perspectives of data, architecture, and learning objectives.

#### 3.2.1 VARIOUS MODALITY-CENTRIC DATA

C-MCR only uses data of overlapping modality to retrieve semantically similar embeddings of other modalities and treats these generated embeddings as pseudo pairs (we call single modality-centric data). However, it is difficult to fully represent one modality with another, and retrieved embeddings by one modality often ignore some semantics of other modalities. For example, image about "mushroom" tend to be absent when retrieving embeddings by audios, and audio of "wind noise" may be ignored in embeddings aggregated by images. Therefore, aggregating embeddings from only a single modality struggles to capture the entire representation space of different modalities.

To tackle the above problem, we propose various modality-centric data strategy. By ensembling semantic consistent embeddings aggregated by multiple modalities, the final embeddings can reflect the representation space of different modalities in different MCRs more comprehensively. As depicted in Fig. 1 (b), all modalities in two spaces are iteratively employed as queries to aggregate corresponding semantic consistent embeddings. Take aligning audio-text space to text-image space as an example, the consistent embeddings based on overlapping modality (e.g., text) are aggregated as follows:

$$\tilde{\mathbf{t}}_i^A = \mathbf{t}_i^A; \quad \tilde{\mathbf{a}}_i^A = \text{softmax}((\tilde{\mathbf{t}}_i^A \cdot \mathbf{T}^A)/\tau_1) \cdot (\mathbf{T}^A)^T$$
$$\tilde{\mathbf{t}}_i^I = \mathbf{t}_i^I; \quad \tilde{\mathbf{v}}_i^I = \text{softmax}((\tilde{\mathbf{t}}_i^I \cdot \mathbf{V}^I)/\tau_1) \cdot (\mathbf{V}^I)^T \tag{1}$$

where the $\tau_1$ is the temperature parameter of softmax, and the softmax is over all the samples in used datasets. The tilde symbols mean the features are processed to be semantic consistent. The $\tilde{\mathbf{t}}_i^A$ and $\tilde{\mathbf{t}}_i^I$ are derived from the same text data, and their semantics are natively consistent. Benefiting from the modality semantic alignment within each pre-trained space, the generated $\tilde{\mathbf{a}}_i^A$ and $\tilde{\mathbf{v}}_i^I$ are also semantically relevant to the $\tilde{\mathbf{t}}_i^A$ and $\tilde{\mathbf{t}}_i^I$.

To capture the representation space of non-overlapping modality more comprehensively, we further aggregate semantic consistent embeddings via data of non-overlapping modality (e.g., audio and image). The process of generating embeddings based on audio can be expressed as:

$$\tilde{\mathbf{a}}_i^A = \mathbf{a}_i^A; \quad \tilde{\mathbf{t}}_i^A = \text{softmax}((\tilde{\mathbf{a}}_i^A \cdot \mathbf{T}^A)/\tau_1) \cdot (\mathbf{T}^A)^T$$
$$\tilde{\mathbf{t}}_i^I = \text{softmax}((\tilde{\mathbf{a}}_i^A \cdot \mathbf{T}^A)/\tau_1) \cdot (\mathbf{T}^I)^T; \quad \tilde{\mathbf{v}}_i^I = \text{softmax}((\tilde{\mathbf{t}}_i^I \cdot \mathbf{V}^I)/\tau_1) \cdot (\mathbf{V}^I)^T \tag{2}$$

Since the embeddings of $\mathbf{T}^A$ and $\mathbf{T}^I$ of overlapping modality are one-to-one matched, the similarity weights between $\tilde{\mathbf{a}}_i^A$ and $\mathbf{T}^A$ can be naturally transferred to $\mathbf{T}^I$.

Based on aforementioned formulas, when extending audio-text to text-image, we iteratively employ texts, audios and images as queries to aggregate corresponding semantic consistent embeddings. During training, semantic consistent embeddings from different sources are shuffled together and the final data pool of various modality-centric data can be represented as $\{\tilde{\mathbf{a}}_i^A, \tilde{\mathbf{t}}_i^A, \tilde{\mathbf{t}}_i^I, \tilde{\mathbf{v}}_i^I\}_{i=0}^n$.

### 3.2.2 DECOUPLED PROJECTOR

The main network structure of Ex-MCR is a projector, and it serves two purposes: 1) Learning the intra-space alignment to close the modality gaps within leaf-MCR and prompt more stable alignment between spaces. 2) Learning the inter-space alignment for extending leaf-MCR to base-MCR. Considering these two different purposes, we propose a decoupled projector to alleviate the potential conflict between distinct optimization objectives and explore a more reasonable mapping layer design for these two purposes. As shown in Fig. 1, the projector is decoupled into a linear layer $f_l(\cdot)$ for intra-space alignment and a multi-layer perceptron layer $f_m(\cdot)$ for inter-space alignment. For extending CLAP to CLIP, we first use $f_l$ to align $\tilde{\mathbf{a}}_i^A$ to $\tilde{\mathbf{t}}_i^A$, the loss function is defined as:

$$L_{intra} = \frac{1}{2}\frac{1}{B}\sum_{i=1}^B \|f_l(\tilde{\mathbf{a}}_i^A) - \tilde{\mathbf{t}}_i^A)\|_2 \tag{3}$$

With the intra-space alignment loss, $f_l(\cdot)$ learns the mapping between audio subspace and text subspace within the CLAP, thereby effectively closing the modality gap. Since the subspaces of different modalities within pre-trained spaces are actually very similar, linear mapping is enough to bridge the modality gap. Moreover, our experiments even found that activation layers have a negative effect on bridging the modality gap.

After bridging the modality gap, the shared $f_m(\cdot)$ are employed to map both audio and text embeddings of CLAP space to the CLIP space, which can be expressed as:

$$\hat{\mathbf{a}}_i^A = f_m(f_l(\tilde{\mathbf{a}}_i^A)); \quad \hat{\mathbf{t}}_i^A = f_m(\tilde{\mathbf{t}}_i^A) \tag{4}$$

### 3.2.3 DENSE ALIGNMENT OBJECTIVE

Since the modality gap within base-MCR is still preserved, a more robust learning objective is needed to map leaf-MCR to the appropriate position in the base-MCR space. To this end, we propose

to learn the alignment densely among the quadruple semantic consistent embedding pairs described in Sec. 3.2.1. When extending CLAP to CLIP, the dense inter-space alignment losses are defined as:

$$L_{avc} = \text{InfoNCE}(\hat{\mathbf{a}}^A, \tilde{\mathbf{v}}^I); \quad L_{tvc} = \text{InfoNCE}(\hat{\mathbf{t}}^A, \tilde{\mathbf{v}}^I)$$
$$L_{atc} = \text{InfoNCE}(\hat{\mathbf{a}}^A, \tilde{\mathbf{t}}^I); \quad L_{ttc} = \text{InfoNCE}(\hat{\mathbf{t}}^A, \tilde{\mathbf{t}}^I) \tag{5}$$

where the $\text{InfoNCE}(\cdot, \cdot)$ is the standard contrastive loss function, which is defined as:

$$\text{InfoNCE}(\mathbf{x}, \mathbf{z}) = -\frac{1}{2}\frac{1}{B}\sum_{i=1}^{B}\left[\log \frac{\exp(\text{sim}(\mathbf{x}_i, \mathbf{z}_i)/\tau_2)}{\sum_{j=1}^{B}\exp(\text{sim}(\mathbf{x}_i, \mathbf{z}_j)/\tau_2)} + \log \frac{\exp(\text{sim}(\mathbf{z}_i, \mathbf{x}_j)/\tau_2)}{\sum_{j=1}^{B}\exp(\text{sim}(\mathbf{z}_i, \mathbf{x}_j)/\tau_2)}\right] \tag{6}$$

where the $\tau_2$ is the temperature parameter. The overall loss is defined as a weighted combination of the intra-space and inter-space losses:

$$L = \lambda L_{intra} + \frac{1}{4}(L_{avc} + L_{atc} + L_{tvc} + L_{ttc}) \tag{7}$$

where $\lambda$ is the hyper-parameter to balance the two terms.

Various modality-centric data 3.2.1, decoupled projector 3.2.2, and dense alignment loss 3.2.3 are also symmetrically employed to extend the 3D-image space to image-text space via images. As a result, we obtain a unified 3D-image-text-audio representation. Considering audio, text, image, and 3D point cloud inputs, we use CLAP's audio encoder, CLIP's text and image encoder, and ULIP's 3D encoder to extract corresponding features $\mathbf{a}_i^A$, $\mathbf{t}_i^I$, $\mathbf{v}_i^I$, $\mathbf{p}_i^U$. The $\mathbf{t}_i^I$, $\mathbf{v}_i^I$, $f_m^A(f_l^A(\mathbf{a}_i^A))$, $f_m^U(f_l^U(\mathbf{p}_i^U))$ are the final audio-text-image-3D unified representation learned by Ex-MCR, where the $f_m^A(\cdot), f_l^A(\cdot); f_m^U(\cdot), f_l^U(\cdot)$ are the learned projectors of CLAP and ULIP respectively.

## 4 EXPERIMENT

### 4.1 EXPERIMENTAL SETTING

**Datasets** For a fair comparison, we use the same unimodal datasets to C-MCR (Wang et al., 2023b) for training, totaling 2.31M texts, 1.3M images, 1.8M audio, and 0.8M 3D point clouds. More details about training datasets are provided in the Appendix.

**Implementation Details** We employ pre-trained frozen CLIP ViT-B/32 (Radford et al., 2021), CLAP (Wu et al., 2023), and ULIP v2 (PointBERT version) (Xue et al., 2023b) models. The temperature $\tau_1$ in Eq. 1 2 for embedding aggregation is set to 0.01 following Wang et al. (2023b), while the $\tau_2$ in 6 is set to 0.05. The hyper-parameter $\lambda$ in Eq. 7 is set to 0.1. Following Wang et al. (2023b), we also add Gaussian noise with a variance of 0.004 to the semantic consistent embeddings described in Sec. 3.2.1. The linear projector $f_l(\cdot)$ is a simple linear layer, and the MLP projector $f_m(\cdot)$ is a 2-layer MLP. We train our model with a batch size of 4096 for 36 epochs. We employ the AdamW optimizer with an initial learning rate of 1e-3 and a cosine learning rate decay strategy.

### 4.2 AUDIO-IMAGE-TEXT RESULTS

**Downstream Tasks** We employ zero-shot audio-image, audio-text, and image-text retrieval tasks to evaluate the audio-image-text representations of Ex-MCR. For audio-image retrieval, we conduct evaluations on Flickr-SoundNet (Senocak et al., 2018), VGGSS (Chen et al., 2021), and AVE (Tian et al., 2018) datasets. Due to their small dataset sizes, we utilize all their available data, comprising 5,000, 5,000, and 4,097 samples. For audio-text retrieval, we utilize the validation set from the AudioCaps (Kim et al., 2019) dataset, which includes 964 audio samples, and for each audio, we choose one corresponding caption for retrieval. Regarding image-text retrieval, we employ the validation set of COCO (Lin et al., 2014) dataset, consisting of 5,000 images, each accompanied by text captions. We randomly select one text annotation for each image as the ground truth. We calculate the cosine similarity between modalities in representation space and use mAP and Top-5 metrics for performance comparison.

Table 1: Results of audio-image-text experiments. The best results are **bolded**. WAV2CLIP is trained on VGG-Sound. Its retrieval results on VGGSS are supervised, while other methods are zero-shot. Therefore, the results of WAV2CLIP on VGGSS are marked gray.

| Method | Audio-Image FlickrNet | | Audio-Image AVE | | Audio-Image VGGSS | | Audio-Text AudioCaps | | Image-Text COCO | |
|---|---|---|---|---|---|---|---|---|---|---|
| | mAP | R@5 | mAP | R@5 | mAP | R@5 | mAP | R@5 | mAP | R@5 |
| CLAP | - | - | - | - | - | - | 21.98 | 35.23 | - | - |
| CLIP | - | - | - | - | - | - | - | - | 44.57 | 57.62 |
| AudioCLIP | 3.81 | 4.91 | 2.33 | 2.65 | 3.10 | 3.94 | 2.23 | 2.68 | 20.14 | 27.42 |
| WAV2CLIP | 2.77 | 3.41 | 3.48 | 4.23 | 7.42 | 10.47 | 0.88 | 0.99 | 44.57 | 57.62 |
| C-MCR | 4.74 | **5.97** | 4.21 | 4.91 | 5.95 | 7.69 | 9.50 | 13.62 | 24.56 | 33.83 |
| Ex-MCR | **4.94** | 5.95 | **4.46** | **4.93** | **6.39** | **8.12** | **11.19** | **16.65** | **44.57** | **57.62** |

Table 2: Results of 3D-image-text experiments.

| Method | 3D-Text ModelNet40 | | | 3D-Image Objaverse-LVIS | | | Image-Text COCO | | |
|---|---|---|---|---|---|---|---|---|---|
| | Acc@1 | Acc@3 | Acc@5 | mAP | R@1 | R@5 | mAP | R@1 | R@5 |
| CLIP | - | - | - | - | - | - | 44.57 | 32.58 | 57.62 |
| ULIP | 60.40 | 79.00 | 84.40 | 3.54 | 1.45 | 4.51 | 34.42 | 22.92 | 46.33 |
| ULIP v2 | **73.06** | 86.39 | 91.50 | **11.41** | **6.00** | **15.63** | 34.42 | 22.92 | 46.33 |
| C-MCR | 64.90 | 87.00 | 92.80 | 3.84 | 1.36 | 4.80 | 24.23 | 14.34 | 33.19 |
| Ex-MCR | 66.53 | **87.88** | **93.60** | 6.23 | 2.54 | 8.25 | **44.57** | **32.58** | **57.62** |

**Performance Comparison**   Fig. 1 compares Ex-MCR with WAV2CLIP, AudioCLIP, and C-MCR. Notably, even without using audio-image paired data, Ex-MCR achieves significantly better performance over WAV2CLIP and AudioCLIP, which illustrates that Ex-MCR is a more effective representation learning method when high-quality data pairs are limited. Furthermore, compared to C-MCR, Ex-MCR not only attains better audio-image alignment but also inherits more audio-text alignment from CLAP, with fully preserved image-text modality alignment of CLIP, which demonstrates the overall superiority of Ex-MCR over C-MCR in establishing new spaces and maintaining original spaces. In summary, extending CLAP to CLIP with our Ex-MCR method derives state-of-the-art audio-image-text unified representations.

## 4.3   3D-IMAGE-TEXT RESULTS

**Downstream Tasks**   To evaluate the performance of 3D-image-text space learned by extending ULIP to CLIP, we conduct a zero-shot 3D object classification task to assess the alignment between 3D and text. We also perform zero-shot 3D-image and image-text retrieval tasks to evaluate the alignment between 3D and image, as well as image and text. The zero-shot 3D object classification task is carried on the ModelNet40 (Wu et al., 2015) validation set, and we use the same prompt strategy as Xue et al. (2023b). Regarding the zero-shot 3D-image retrieval task, we use the Objaverse-LVIS dataset (Deitke et al., 2023), which includes 46,054 3D objects. Additionally, we continued to use the COCO dataset's validation set for zero-shot image-text retrieval.

**Performance Comparison**   It is worth noting that ULIP aligns a 3D encoder to a vision-language model called SLIP (Mu et al., 2022) (not CLIP) through 3D-image-text data. Ex-MCR only uses the aligned 3D-image representation of ULIP to extend it to a different vision-language model (i.e., CLIP) via the paired-data-free way. So we are not reproducing or refining the alignment of ULIP, but building a new alignment from scratch between the 3D representation of ULIP and CLIP. From Tab. 4.3, we can find the following key points. Firstly, even without using any 3D-text data, Ex-MCR still outperforms the advanced models (ULIP and ULIP v2) trained on 3D-text pairs in most performance metrics for 3D object classification. Secondly, the 3D-image retrieval accuracy of Ex-MCR is significantly higher than ULIP and C-MCR but lower than ULIP v2. Since the 3D-image space of ULIP v2 is treated as leaf-MCR, it is reasonable that Ex-MCR 3D-image performance is slightly lower than ULIP v2. At the same time, the better 3D-image retrieval accuracy than ULIP and

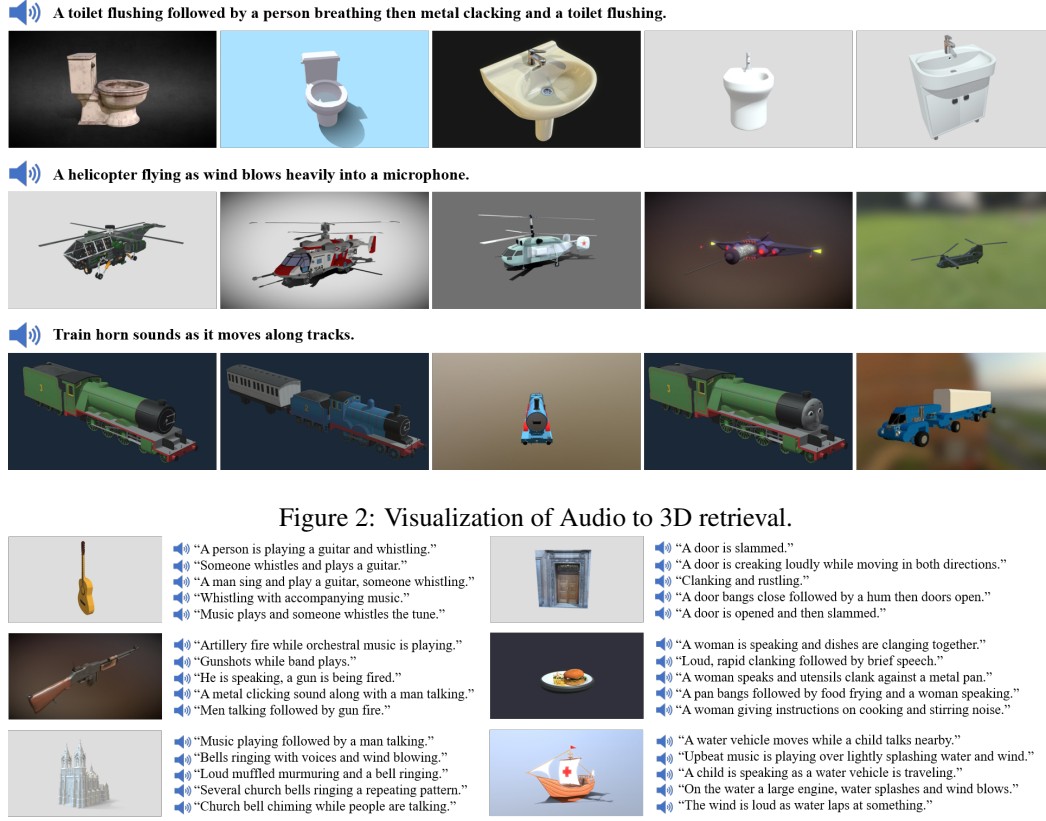

Figure 2: Visualization of Audio to 3D retrieval.

Figure 3: Visualization of 3D to Audio retrieval.

C-MCR shows that Ex-MCR effectively learns strong 3D-image alignment. Finally, Ex-MCR retains the best image-text retrieval accuracy compared to these previous state-of-the-art models. The leading performance on all these tasks further demonstrates the superiority of Ex-MCR in unified contrastive representation learning.

### 4.4 EMERGENT 3D-AUDIO ALIGNMENT

In this section, we study whether the semantic alignment also emerges between the extended modalities (e.g., audio and 3D). We mutually retrieve audio in AudioCaps and 3D objects in Objaverse. In Fig. 2 and 3, we provide visualizations of some top-5 retrieval results, and audios are described by their corresponding caption annotations. These cases effectively demonstrate the emergent semantic alignment between audio-3D in Ex-MCR space. For example, the sound of a flushing toilet and water flow can retrieve 3D objects of toilets or sinks, while a sailboat 3D object can retrieve clips containing sounds of water vessels and wind. More results and the original audio files are provided in our supplementary material.

These exciting results demonstrate that extending ULIP and CLAP onto CLIP following our Ex-MCR methods derives a 3D-vision-text-audio unified contrastive representation space. In addition to the state-of-the-art performance on all possible tasks, Ex-MCR is an extremely training-efficient and paired-data-free representation learning method, which amplifies its application value in unified multi-modal representation learning.

### 4.5 ABLATION STUDIES

In this section, we analyze the main components of Ex-MCR. All experiments are conducted on extending CLAP to CLIP, and we reported the average mAP of audio-visual and audio-text retrieval on AVE and AudioCaps datasets, respectively. In addition, we also provide results on more datasets and evaluation metrics in the Appendix.

Table 3: Various modality-centric data. We report the mAP metrics on audio-image retrieval (AVE) and audio-text retrieval (AudioCaps).

|   | AVE | AudioCaps |
|---|---|---|
| A | 4.10 | 11.11 |
| I | 3.41 | 5.54 |
| T | 4.17 | 9.89 |
| A+I | 4.11 | 11.09 |
| A+T | 4.12 | 10.88 |
| I+T | 4.05 | 8.39 |
| A+I+T | **4.46** | **11.19** |

Table 4: Alignment objective. A-T, T-T, A-V, and T-V represent the alignment objective between audio-text, text-text, audio-image, and text-image, respectively.

|   | AVE | AudioCaps |
|---|---|---|
| A-T | 4.00 | 10.82 |
| T-T | 4.15 | **11.30** |
| A-V | 3.97 | 7.49 |
| T-V | 4.18 | 7.68 |
| All | **4.46** | 11.19 |

Table 5: Structure of $f_1(\cdot)$

| $f_1(\cdot)$ | AVE | AudioCaps |
|---|---|---|
| Linear | **4.46** | **11.19** |
| 1 MLP | 4.16 | 10.25 |
| 2 MLP | 4.04 | 9.93 |

Table 6: Structure of $f_m(\cdot)$

| $f_m(\cdot)$ | AVE | AudioCaps |
|---|---|---|
| Linear | 3.70 | 11.15 |
| 1 MLP | 4.15 | 10.53 |
| 2 MLP | **4.46** | 11.19 |
| 3 MLP | 4.31 | **11.30** |
| 4 MLP | 4.35 | 11.07 |
| 5 MLP | 4.42 | 10.93 |

**Various modality-centric data** As described in Sec. 3.2.1, we employ various modality-centric data to train our projectors. For investigating the effect of different modality-centric data, we ablate each modality-centric data, and the results are reported in Tab. 3. The A, I, and T represent pseudo data derived from audio, image and text respectively. Each kind of data is beneficial for audio-visual and audio-image alignment, and using all kinds of data simultaneously brings the best performance. In addition, we find that pseudo-pairs from audios are critical to the performance of audio-text retrieval, demonstrating the importance of various modality-centric data, and proving that previous single modality-centric data really can not fully reflect the audio representation space.

**Dense alignment objective** To analyze the impact of different alignment objectives, we train the model with each alignment objective. From the results reported in Tab. 4, we find that directly aligning the pseudo audio-image or audio-text embedding pairs leads to sub-optimal audio alignment, whereas aligning spaces by overlapping text modality brings better alignment than learning alignment directly from pseudo pairs. This observation further suggests that overlapping modalities play a key pivotal role in aligning different spaces.

**Structure of $f_l(\cdot)$** Tab. 5 demonstrates the impact of different structures of $f_l(\cdot)$. The results prove our hypothesis: the representation structures between different modalities within one MCR space are similar, and a simple linear layer is enough to bridge the modality gap. Moreover, the activation layer of the MLP introduces non-linearity, which may disrupt the spatial structure of representations.

**Structure of $f_m(\cdot)$** The ablation studies of $f_m(\cdot)$ are summarized in Tab. 6. When aligning different MCR spaces, the nonlinear MLP structure with stronger expressivity is better than the simple linear layer. Besides, good results are achieved no matter how many layers of MLP, which demonstrates the robustness of our method. According to more detailed experiments in Tab. 11, empirically, MLP with 2 or 3 layers achieves a good balance between expressivity and learning difficulty.

## 5  CONCLUSION

This paper proposes **Ex**tending **M**ulti-modal **C**ontrastive **R**epresentations (Ex-MCR), a novel training-efficient and paired-data-free unified constrastive representation learning method for more than three modalities. Ex-MCR effectively integrates the knowledge in pre-trained MCRs through overlapping modalities between these MCRs. By extending ULIP and CLAP onto CLIP via the overlapping image and text modality, respectively, we derive unified and high-quality audio-image-text-3D representations. Without using any paired data, Ex-MCR attains a series of state-of-the-art performance results across various tasks. More importantly, semantic alignment is also observed between extended modalities (e.g., audio-3D), which highlights the potential of Ex-MCR in modality extensibility.

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

# A    TRAINING DATASET

The details of our training dataset, which are mentioned in Sec. 4.1, are shown below.

**Text Dataset**    To ensure that the texts contain sufficient information for other modalities, the data of text is sourced from diverse perspectives in vision-text datasets (COCO, CC3M), video-text datasets (MSRVTT, MAD), and audio-text datasets (AudioCaps, Clotho). Following Wang et al. (2023b), we select 1M texts from CC3M. There are 2.33M text samples in total. We extract their CLAP and CLIP features $\mathbf{T}^A$ and $\mathbf{T}^I$ using the CLAP and CLIP encoders, respectively.

**Image Dataset**    For another modality in base-MCR, Vision, we utilize ImageNet1K as the data source. ImageNet1K is a large-scale image recognition dataset consisting of 1.3 million images. We extract their features to the sets $\mathbf{V}^I$, and $\mathbf{V}^U$ in CLIP and ULIP, using the CLIP Encoder and ULIP Encoder.

**Audio Dataset**    AudioSet is a large-scale audio dataset with 2.1M audio clips from YouTube, equivalent to 5.8 thousand hours of audio and encompassing over 500 sound classes. We use the CLAP audio encoder to extract the feature set $\mathbf{A}^A$ from the audios of the training set.

**3D Point Cloud Dataset**    For the 3D modality, we use Objaverse, the recently released and large-scale 3D objects dataset. It has approximately 800K real-world 3D objects. All 3D data are transformed into point clouds and extracted into the feature set $\mathbf{P}^U$ using the ULIP 3D encoder.

It is worth noting that we do not employ any annotations provided with the datasets mentioned above as part of our training data, which means we only use the unimodal modality of data in each dataset we selected.

# B    ARCHITECTURE OF PROJECTORS

Table 7: Model configurations of projectors.

| Module | Block | $C_{in}$ | $C_{out}$ |
|---|---|---|---|
| $f_1(\cdot)$ | Linear | 512 | 512 |
| | Linear | 512 | 1024 |
| | BatchNorm1D | 1024 | 1024 |
| | Relu | - | - |
| | Linear | 1024 | 512 |
| | BatchNorm1D | 512 | 512 |
| | Relu | - | - |
| $f_m(\cdot)$ | Linear | 512 | 1024 |
| | BatchNorm1D | 1024 | 1024 |
| | Relu | - | - |
| | Linear | 1024 | 512 |
| | BatchNorm1D | 512 | 512 |
| | Relu | - | - |

The model configurations of our projectors are shown in Tab. 7.

# C    DETAILED RESULTS OF ABLATION STUDY

As a supplement to Tab. 3, Tab. 4, Tab. 5, and Tab. 6, we provide detailed ablation experiment results on more comprehensive evaluation metrics of various datasets, as shown below.

Table 8: Detailed results of experiments on data modality-centric.

| Data Perspective | FlickrNet | | AVE | | VGGSS | | AudioCaps | |
|---|---|---|---|---|---|---|---|---|
| | mAP | R@5 | mAP | R@5 | mAP | R@5 | mAP | R@5 |
| A | 3.94 | 4.77 | 4.10 | 4.66 | 5.47 | 6.95 | 11.11 | 16.39 |
| I | 3.83 | 4.63 | 3.41 | 3.70 | 4.82 | 5.96 | 5.54 | 7.18 |
| T | 4.85 | **5.96** | 4.17 | 4.61 | 5.72 | 7.23 | 9.89 | 14.47 |
| A+I | 4.22 | 4.96 | 4.11 | 4.71 | 6.01 | 7.78 | 11.09 | **16.91** |
| A+T | 4.63 | 5.56 | 4.12 | 4.64 | 5.88 | 7.57 | 10.88 | 16.23 |
| I+T | 4.70 | 5.82 | 4.05 | 4.34 | 5.84 | 7.36 | 8.39 | 12.09 |
| A+I+T | **4.94** | 5.95 | **4.46** | **4.93** | **6.39** | **8.12** | **11.19** | 16.65 |

Table 9: Detailed results of experiments on alignment objective.

| Objective | FlickrNet | | AVE | | VGGSS | | AudioCaps | |
|---|---|---|---|---|---|---|---|---|
| | mAP | R@5 | mAP | R@5 | mAP | R@5 | mAP | R@5 |
| A-T | 4.01 | 4.78 | 4.00 | 4.56 | 5.70 | 7.28 | 10.82 | 15.87 |
| T-T | 4.56 | 5.33 | 4.15 | 4.54 | 5.68 | 6.86 | **11.30** | **16.93** |
| A-V | 4.30 | 5.34 | 3.97 | 4.51 | 5.91 | 7.30 | 7.49 | 10.35 |
| T-V | 4.77 | **6.03** | 4.18 | 4.92 | 5.43 | 6.93 | 7.68 | 10.36 |
| Dense | **4.94** | 5.95 | **4.46** | **4.93** | **6.39** | **8.12** | 11.19 | 16.65 |

Table 10: Detailed results of experiments on the structure of $f_1(\cdot)$.

| $f_1(\cdot)$ | FlickrNet | | AVE | | VGGSS | | AudioCaps | |
|---|---|---|---|---|---|---|---|---|
| | mAP | R@5 | mAP | R@5 | mAP | R@5 | mAP | R@5 |
| Linear | **4.94** | **5.95** | **4.46** | **4.93** | 6.39 | 8.12 | **11.19** | **16.65** |
| 1 MLP | 4.54 | 5.59 | 4.16 | 4.75 | **6.50** | **8.54** | 10.25 | 14.92 |
| 2 MLP | 4.36 | 5.15 | 4.04 | 4.66 | 6.00 | 7.63 | 9.93 | 14.48 |

Table 11: Detailed results of experiments on the structure of $f_m(\cdot)$.

| $f_m(\cdot)$ | FlickrNet | | AVE | | VGGSS | | AudioCaps | |
|---|---|---|---|---|---|---|---|---|
| | mAP | R@5 | mAP | R@5 | mAP | R@5 | mAP | R@5 |
| Linear | 3.62 | 4.50 | 3.70 | 4.03 | 5.40 | 6.82 | 11.15 | 16.37 |
| 1 MLP | 4.62 | 5.79 | 4.15 | 4.76 | 5.81 | 7.28 | 10.53 | 15.87 |
| 2 MLP | 4.94 | 5.95 | **4.46** | 4.93 | 6.39 | 8.12 | 11.19 | 16.65 |
| 3 MLP | 4.85 | 5.93 | 4.31 | 4.88 | 6.57 | 8.70 | **11.30** | **17.10** |
| 4 MLP | **4.95** | **6.20** | 4.35 | 4.84 | 6.55 | 8.57 | 11.07 | 16.23 |
| 5 MLP | 4.79 | 6.02 | 4.42 | **5.15** | **6.59** | **8.63** | 10.93 | 16.21 |

