# OpenReview forum: "Extending Multi-modal Contrastive Representations"
_ICLR.cc/2024/Conference — ICLR 2024 Conference Desk Rejected Submission_

### Official Review · Reviewer_GFCr · 2023-10-27

**Soundness:** 2 fair
**Presentation:** 2 fair
**Contribution:** 2 fair
**Rating:** 6
**Confidence:** 4

**Summary:**

This paper proposed a training-efficient and paired-data-free method to flexibly learn unified contrastive representation space for more than three modalities by integrating the knowledge of existing MCR spaces.

**Strengths:**

(1) The paper formulation is good and clear.

(2) The proposed method can align multiple existing MCRs into the same based MCR, which can effectively preserve the original semantic
alignment of the based MCR.

**Weaknesses:**

(1) The reviewer finds the presentation of figures confusing. For instance, in Table 1, superior results are subtly highlighted in grey to minimize attention, while the results of the proposed model are emphasized in bold to draw more focus.

(2) In Table 3, the value 11.19 is highlighted in bold instead of 11.30.

(3) Considering the issues raised earlier, the reviewer remains uncertain about the authenticity and reproducibility of the results.

(4) Apart from presenting numerical data, what are the additional findings and conclusions drawn from the ablation studies?

**Questions:**

Please see the comments above.

---

> ### Author Response · Authors · 2023-11-20
> **Response to Reviewer GFCr**
>
> ### **W1: grey mark in Table 1 and Table 2:**
>
> A1: CLIP and CLAP are used as base-MCR and leaf-MCR, and they only focus on contrastive representation between two modalities (i.e., image-text and audio-text). Our experiments mainly focus on comparing the unified representations for three or more modalities over various cross-modal downstream tasks rather than the performance on specific image-text or audio-text tasks. The results of CLIP and CLAP are listed to compare how well C-MCR and Ex-MCR inherit the semantic alignment of original MCR spaces. Moreover, in the responses to reviewer 6GyA’s W1 & Q1 & Q3, we discuss in detail why Ex-MCR’s performance on audio-text or 3D-image lags behind that of the leaf-MCR is acceptable and some future directions to further improve performance.
>
> In Table 1, the baseline WAV2CLIP is trained on VGG-Sound, and the evaluation dataset VGGSS is a subset of the VGG-Sound test set. This means the evaluation of WAV2CLIP on VGGSS is supervised. However, other methods do not use any VGG-sound data for training, so other results on VGGSS are evaluated in a zero-shot manner. Therefore, we grayed out the results of WAV2CLIP on VGGSS.
>
> Sorry for the misunderstanding caused by not clearly explaining the gray markings, and we have added these explanations in the newly submitted version.
>
> ### **W2: 11.30 and 11.19 in Table 3**
>
> A2: I suppose you are probably referring to 11.30 and 11.19 in Table 4. Sorry for the misunderstanding, this is a typo, 11.30 should be bolded, not 11.19. We are not intentionally bolding it incorrectly. In the similar case in Table 6, the marks are correct.
>
> Moreover, in Table 11, we display more ablation results about alignment objectives. Using all alignment objectives shows obvious advantages over most tasks and metrics. This incorrect bold mark would not affect the conclusion that the dense alignment objective is effective.
>
> Thank you for your kind reminder. We have modified this typo in the newly submitted paper.
>
> ### **W3 Considering the issues raised earlier, the reviewer remains uncertain about the authenticity and reproducibility of the results.**
>
> A3: The gray marks in Tables 1 and 2 are explained in the response to W1. For W2, it is a typo and it has been modified. The detailed ablation experimental results are also shown in the appendix, further supporting our conclusions.
>
> To completely eliminate concerns about the authenticity and reproducibility of the results, we provide Ex-MCR pre-trained weights, well-packaged code, download links for the pre-trained models used, and guidelines for using our Ex-MCR in the new submission's supplementary materials for reproduction. These pre-trained weights and code will be open source.
>
> ### **W4 Additional findings and conclusions drawn from the ablation studies?**
>
> For ablation results on “Various modality-centric data”, we further find that pseudo-pairs from audios are critical to the performance of audio-text retrieval, demonstrating the importance of various modality-centric data, and proving that previous single modality-centric data really can not fully reflect the audio representation space.
>
> For ablation results on “Dense alignment objective”, we find that directly aligning the pseudo audio-image or audio-text embedding pairs leads to sub-optimal audio alignment, whereas aligning spaces by overlapping text modality brings better alignment than learning alignment directly from pseudo pairs. This observation further suggests that overlapping modalities play a key pivotal role in aligning different spaces.
>
> For ablation results on “Structure of $f_m(\cdot)$”, we summarize that good results are achieved no matter how many layers of MLP, which demonstrates the robustness of our method. According to more detailed experiments, empirically, MLP with 2 or 3 layers achieves a good balance between expressivity and learning difficulty.

---

> ### Author Response · Authors · 2023-11-22
> **Deadline coming. Looking forward to your feedback.**
>
> Dear Reviewer GFCr,
>
> Thanks again for your comments. We would like to kindly remind you that we tried our best to address the concerns you raised.
>
> We kindly request your feedback as the rebuttal deadline is approaching in less than 1 day. We would be happy to discuss in detail if you have additional comments about our paper.
>
> Best regards, Authors

---

> > ### Comment · Reviewer_GFCr · 2023-12-01
> > **Official Comment by Reviewer GFCr**
> >
> > I appreciate the authors for their rebuttal. I am inclined to trust the author's assertion that the misrepresentation of results was not deliberate, and as a result, I have adjusted my score accordingly, as the method and results look promising now. However, I also agree with the other reviewers that there is room for improvement in terms of the overall clarity and writing of the paper, I would like to encourage the authors to consider revising it further in accordance with the feedback provided by other reviewers.

---

### Official Review · Reviewer_CZb4 · 2023-11-01

**Soundness:** 3 good
**Presentation:** 2 fair
**Contribution:** 3 good
**Rating:** 6
**Confidence:** 5

**Summary:**

this paper introduces Ex-MCR, a method for learning unified contrastive representations across more than three modalities, focusing on being training-efficient and not requiring paired data

**Strengths:**

the idea of extending multi-modal contrastive representation (Ex-MCR) to incorporate more than three modalities without relying on paired data is an interesting and important topic given there are appearing more multimodal foundation models, but each of them only covers a fraction of the modalities.
The proposed approach is technically sound.

**Weaknesses:**

one of the concerns here is the baselines and the results, even the methods like imagebind use paired data, the reviewer still thinks that it's worthwhile to study what will be the performance differences, as it could indicate the limitation/boundary of this kind of paired-data-free methods.

**Questions:**

minor issues:
typos in the first summarized contribution in the introduction.
figure 1 seems to lack some clarity, and the caption is not very helpful for understanding this figure, consider improving this part.

---

> ### Author Response · Authors · 2023-11-20
> **Response to Reviewer CZb4**
>
> ### **W1: the imagebind baselines & the limitation/boundary of this kind of paired-data-free methods**
>
> **Comparison with imagebind:**
>
> The current compared baselines: AudioCLIP, WAV2CLIP and ULIP, ULIP v2. These methods are all trained on paired audio-image-text or 3D-image-text data, and their model sizes are equivalent to Ex-MCR. Thus, the main difference between Ex-MCR and these baselines is the training data and training strategy. Comparison with these baselines better demonstrates the strengths of our “extending” learning scheme.
>
> However, imagebind not only uses paired data for training, but also uses a much stronger pre-trained model, (Imagebind uses OpenCLIP ViT-Huge trained on LAION-2B while Ex-MCR uses CLIP ViT-Base trained on WIT-400M). Considering the audio-image-text representation, the parameters number of imagebind are 630M (image encoder), 303M (text encoder), and 86M (audio encoder), while the parameters number of our Ex-MCR is 88M (image encoder), 63M (text encoder) and 32M (audio encoder).
>
> Although imagebind uses both data pairs and stronger pre-trained models, we still provide a performance comparison on audio-image-text as:
>
> |                             | | FlickrNet | AVE  | VGGSS | Audiocaps | COCO |
> | -------------------------- |--| --------- | ---- | ----- | --------- | ---- |
> |                            | Total params | R@5       | R@5  | R@5   | R@5       | R@5  |
> | ImageBind (OpenCLIP ViT-H) | 1,019M | **20.78** | **40.11** | **35.67** | 8.29 | **74.29** |
> | Ex-MCR (CLIP ViT-B)        | 183M | 5.95 | 4.93 | 8.12 | 16.65 | 57.62 |
> | Ex-MCR (OpenCLIP ViT-H)    | 965M | 6.35 | 6.90 | 11.59 | **25.00** | **74.29** |
>
> On the image-audio task, imagebind achieves better results since it uses large-scale paired data and stronger visual pre-training. On the audio-text task, our method is far better than imagebind. The performance difference in image-text tasks is due to using the different CLIP models.
>
> **The potential of this kind of paired-data-free methods**
>
> In the responses to Reviewer 6GyA, we fine-tune the projector with paired data which brings performance improvements. Besides, we further discuss the potential of Ex-MCR in scalability and low computing resource situations.
>
> Regarding the boundaries and potential of absolute performance, we try to replace the CLIP ViT-Base with the larger OpenCLIP ViT-Huge. The results are also reported in the above table. Simply using a larger version of CLIP, the performances on audio-image, audio-text, and image-text tasks are significantly improved.
>
> In addition, there are many future directions to further improve the absolute performance of Ex-MCR, such as using more unimodal data, integrating paired data into the training process, employing stronger leaf-MCR pre-training, and designing more sophisticated space alignment architecture and learning objectives. Our future work will be dedicated to further exploring the potential of such paired-data-free methods.
>
> ### **Q1: Typos. Unclear Figure 1 and its caption.**
>
> Thanks for pointing out the typos and providing suggestions about figures, we have fixed them in the newly submitted paper.
>
> Additionally, in our response to reviewer pnPG, we analyzed most of the modifications that were made to improve the clarity and readability of the presentation.

---

> ### Author Response · Authors · 2023-11-22
> **Deadline coming. Looking forward to your feedback.**
>
> Dear Reviewer CZb4,
>
> Thanks again for your comments. We would like to kindly remind you that we tried our best to address the concerns you raised.
>
> We kindly request your feedback as the rebuttal deadline is approaching in less than 1 day. We would be happy to discuss in detail if you have additional comments about our paper.
>
> Best regards, Authors

---

> > ### Comment · Reviewer_CZb4 · 2023-11-23
> >
> > Thank you for the response. I've read other reviews and rebuttals.

---

### Official Review · Reviewer_6GyA · 2023-11-01

**Soundness:** 3 good
**Presentation:** 4 excellent
**Contribution:** 3 good
**Rating:** 6
**Confidence:** 4

**Summary:**

This paper proposes a novel method, Extending Multimodal Contrastive Representation (Ex-MCR), to address challenges in multi-modal learning for more than three modalities. Traditional methods are constrained by the need for large-scale, high-quality paired data and high training costs. Ex-MCR offers a training-efficient solution that doesn't rely on paired data, by aligning multiple existing MCRs into a base MCR, preserving their original semantic alignment. The method enhances the alignment process through various techniques, including modality-centric pseudo data pairs and a decoupled projector. Experiments demonstrate its state-of-the-art performance on various tasks, showcasing its potential in representation learning and modality extensibility.

**Strengths:**

1. The paper introduces Ex-MCR, which extends one MCR space (leaf-MCR) into another fixed MCR space (base-MCR). This approach optimizes the preservation of modality alignment within the base MCR, showcasing its potential for integrating multiple MCRs.
2. This method operates without the need for paired data, demonstrating its potential for scalability.
3. This method employs a dense contrastive loss on pseudo-pairs between all possible modalities, enhancing the learned alignment's stability.
4. Ex-MCR achieves competitive performance across various zero-shot tasks, even better than strong baselines trained by paired data.

**Weaknesses:**

1. While Ex-MCR outperforms baseline methods, its performance lags behind that of the leaf-MCR.
2. The authors highlight the scalability of Ex-MCR; however, it's contingent upon the new MCR having a modality already present in the base MCR. Otherwise, biases or errors may be exacerbated.

**Questions:**

1. How do you explain the performance gap between Ex-MCR and the original leaf-MCR? Are there specific challenges or limitations inherent to Ex-MCR that contribute to this disparity?
2. Given the premise that a new MCR should have a modality already present in the base MCR for effective expansion, how does Ex-MCR handle scenarios where entirely new modalities (or only modalities in leaf-MCR) need to be integrated?
3. Have you experimented with incorporating paired data into the training process for Ex-MCR? If so, how did it impact the results?
4. If the configuration were altered such that CLAP served as the base-MCR and both CLIP and ULIP were treated as leaf-MCRs, would the performance outcomes remain favorable?
5. How many resources do you use for the training?

---

> ### Author Response · Authors · 2023-11-20
> **Response to Reviewer 6GyA (1/2)**
>
> ### **W1 & Q1 & Q3: why Ex-MCR’s performance lags behind that of the leaf-MCR and possible solutions.**
>
> **1) Taking extending CLAP to CLIP as an example, if you mean why the projected CLAP's audio-text representations lag behind the original CLAP audio-text representations:**
>
> In order to avoid misunderstanding, we further clarify that the final unified Ex-MCR representations are composed of CLIP's image, text representations, projected CLAP's audio representations, and projected ULIP's 3D representations (i.e., the $\mathbf{t}^I_i$, $\mathbf{v}^I_i$, $f_m^A(f_l^A(\mathbf{a}^A_i))$, $f_m^U(f_l^U(\mathbf{p}^U_i))$ are the final audio-text-image-3D unified representation). All our experimental results are based on these four unified aligned representations. Therefore, the audio-text results are evaluated by the projected CLAP's audio and CLIP text representations (i.e., the $\mathbf{t}^I_i$, $f_m^A(f_l^A(\mathbf{a}^A_i))$ ), rather than the projected CLAP's audio-text representations (i.e.,  the $f_m^A(\mathbf{t}^A_i)$, $f_m^A(f_l^A(\mathbf{a}^A_i))$ ).
>
> **2) Taking extending CLAP to CLIP as an example, if you mean why the newly aligned audio-text representation in Ex-MCR's unified space lags behind the CLAP audio-text representations:**
>
> The following are reasons and possible solutions for performance lags behind leaf-MCR.
>
> **Reason for performance lags behind leaf-MCR:**
>
> We think that the performance of Ex-MCR lags behind that of leaf-MCRs for three reasons:
>
>  1、Unchanged features extraction capability. The final Ex-MCR's audio representation is only a transformation of the CLAP's audio representation, and the ability to extract meaningful features from the original data is unchanged.
>
> 2、Non-optimal target space. The representation distribution in fixed base-MCR may not fully distinguish the modalities of leaf-MCRs. For example, “people are talking on the beach” and “people are talking in the living room” are different in CLIP’s text representation, but are similar in the audio-text domain.
>
> 3、Limited training data. The pseudo data we use to extend leaf-MCR to base-MCR are mainly derived from the inherent alignment of leaf-MCR and base-MCR. Therefore, this extending process can be regarded as distilling leaf-MCR’s alignment knowledge to construct a new alignment for base-MCR.
>
> Moreover, Ex-MCR is designed to learn unified representations for more than three modalities rather than for only two modalities. And current experiments in the paper aim to demonstrate the potential of Ex-MCR in the extreme situation of no data pairs can be used. Considering the above facts, we think it is reasonable and acceptable that the current Ex-MCR lags behind leaf-MCR in performance on downstream tasks in leaf-MCR’s specific modality.
>
> **Possible solutions for performance lags behind leaf-MCR:**
>
> Combining the above analyses of limitations, there are many possible solutions to further improve the performance of Ex-MCR. Such as using more unimodal data, integrating paired data used by leaf-MCR into the training process, and employing stronger leaf-MCR and base-MCR pre-trainings.
>
> As you suggested in Q3, we try to integrate the paired data used by leaf-MCR. With limited rebuttal time, we simply use part of CLAP’s training audio-text data (audiocaps training set, and 200k pairs of laion630k) to fine-tune the projector. The results are as follows:
>
> |                    | FlickrNet | AVE      | VGGSS    | Audiocaps |
> | ------------------ | --------- | -------- | -------- | --------- |
> |                    | R@5       | R@5      | R@5      | R@5       |
> | CLAP               | -         | -        | -        | 35.23     |
> | C-MCR              | 5.97      | 4.91     | 7.69     | 13.62     |
> | Ex-MCR             | 5.95      | 4.93     | **8.12** | 16.65     |
> | Ex-MCR (fine-tune) | **6.09**  | **5.03** | 8.00     | **21.11** |
>
> By tuning the learned projector with audio-text data pairs, the performance of audio-text is greatly improved without sacrificing the audio-image performance. This demonstrates the compatibility of our structure with paired data, and paired data can further boost the final performance.
>
> We also evaluate using a larger base-MCR pre-training, which also brings performance improvement. Please refer to the response to Reviewer CZb4's W1.

---

> ### Author Response · Authors · 2023-11-20
> **Response to Reviewer 6GyA (2/2)**
>
> ### **W2 & Q2 & Q4: The new MCR should have a modality already present in the base MCR for effective expansion. And the results of extending new MCR via leaf-MCR.**
>
> As discussed in recent work imagebind [1] and languagebind [2], most existing modalities can be bound to either images or language, and most existing MCRs contain either images or language. From a practical perspective, choosing the powerful CLIP as base-MCR can cover almost all scenarios.
>
> *[1] ImageBind: One Embedding Space To Bind Them All. Rohit Girdhar, Alaaeldin El-Nouby, Zhuang Liu, Mannat Singh, Kalyan Vasudev Alwala, Armand Joulin, Ishan Misra. CVPR2023*
>
> *[2] LanguageBind: Extending Video-Language Pretraining to N-modality by Language-based Semantic Alignment. Bin Zhu, Bin Lin, Munan Ning, Yang Yan, JiaXi Cui, Hongfa Wang, et al. Arxiv2023*
>
> However, to further explore the potential of Ex-MCR, we still try to expand a leaf-MCR through the overlapping modality of another leaf-MCR, **like a chain structure**. As you mentioned in Q4, we regard CLAP as base-MCR and CLIP as leaf-MCR, the audio-image-text results are：
>
> |                                  | FlickrNet | AVE      | VGGSS    | Audiocaps | COCO      |
> | -------------------------------- | --------- | -------- | -------- | --------- | --------- |
> |                                  | R@5       | R@5      | R@5      | R@5       | R@5       |
> | Ex-MCR (Original Tree Structure) | **5.59**  | 4.93     | **8.12** | 16.65     | **57.62** |
> | Ex-MCR (Chain Structure)         | 5.15      | **5.08** | 7.18     | **35.23** | 16.30     |
>
> When using CLAP as base-MCR, the audio-text retrieval accuracy on AudioCaps is much higher than using CLIP as base-MCR. Correspondingly, its image-text retrieval accuracy on COCO is lower than using CLIP as base-MCR. For audio image retrieval, the performance of both variants is comparable.
>
> The most important experiment: use ULIP as leaf-MCR of leaf-MCR, which means indirectly aligning the 3D representation of ULIP to the text representation of CLAP through CLIP. The final 3D-image-text results are:
>
> |                          | ModelNet40 (3D-Text) |           |           | Objaverse-LVIS (3D-Image) |          |           |
> | ------------------------ | -------------------- | --------- | --------- | ------------------------- | -------- | --------- |
> |                          | Acc@1                | Acc@3     | Acc@5     | mAP                       | R@1      | R@5       |
> | ULIP                     | 60.40                | 79.00     | 64.40     | 3.54                      | 1.45     | 4.51      |
> | ULIP v2                  | **73.06**            | 86.39     | 91.50     | **11.41**                 | **6.00** | **15.64** |
> | C-MCR                    | 64.90                | 87.00     | 92.80     | 3.84                      | 1.36     | 4.80      |
> | Ex-MCR (Tree Structure)  | 66.53                | **87.88** | **93.60** | 6.23                      | 2.54     | 8.25      |
> | Ex-MCR (Chain Structure) | 48.82                | 64.75     | 73.99     | 2.26                      | 0.77     | 3.09      |
>
>  The final 3D-text-image performance is indeed lower than directly using ULIP as a leaf-MCR and using CLIP as base-MCR. But it still achieved performance comparable to most baselines, which shows the error accumulation of chain structure is within the acceptable range.
>
> ### **Q5: Training resources:**
>
> In our implementation, extending CLAP into CLIP only requires 4G GPU memory. When training on an A100, it takes 6 hours to converge. Due to its extremely low GPU memory requirements, users can train Ex-MCR on almost any device. We also try to reduce the batch size, and when only 2GB GPU memory is used (of course requiring a longer training time), there is almost no performance degradation.
>
> Our training is extremely resource-efficient. Previous contrastive representation learning methods need to set a larger batch size for training stability, which results in very high GPU memory usage. However, in our method, all pre-trained encoders are frozen, so all features can be pre-extracted and saved offline, and only a linear and MLP layer-based projector is learnable, which greatly reduces the GPU memory cost when increasing the batch size.

---

> ### Author Response · Authors · 2023-11-22
> **Deadline coming. Looking forward to your feedback.**
>
> Dear Reviewer 6GyA,
>
> Thanks again for your comments. We would like to kindly remind you that we tried our best to address the concerns you raised.
>
> We kindly request your feedback as the rebuttal deadline is approaching in less than 1 day. We would be happy to discuss in detail if you have additional comments about our paper.
>
> Best regards, Authors

---

### Official Review · Reviewer_pnPG · 2023-11-01

**Soundness:** 3 good
**Presentation:** 2 fair
**Contribution:** 3 good
**Rating:** 6
**Confidence:** 3

**Summary:**

The work proposes an effective and efficient recipe to integrate multiple Multi-modal Contrastive Representation (MCR) spaces into one, making it possible to have a 3D-Text-Vision-Audio model without requiring any additional paired data. This is achieved by leveraging the modality that is shared across the multiple spaces, learning to project each source space (called leaf-MCR) to a chosen target space (base-MCR) by using a InfoNCE loss over pseudo-pairs containing the different modality combinations. To further improve the alignment, the intra-MCR modality gap is closed with an additional regularizing term. The experiments show improved results when considering the union of more than 3 modalities (3D, Text, Vision, Audio), evaluating the performance on cross-modal retrieval tasks over all the possible combinations.

**Strengths:**

I find the paper to offer a significant contribution in multi-modal learning, proposing an efficient way to achieve a multi-modal model on more than 3 modalities without the need for a multi-modal paired dataset annotated for all the considered modalities. As a possible solution to overcome the need for large-scale multi-modal paired datasets, the approach is advisable and useful for the community. In the current AI environment, it is good to see that there is still a strive to push for more accessible approaches that don’t require extensive resources.

### Novelty

- The work goes to build upon C-MCR, inheriting its benefits and effectively overcoming its limitations, in practice extending its applicability to multiple modalities altogether.

### Quality

- The methods employed are simple yet effective, and the overall framework is rather lightweight, making it directly usable by any practitioner desiring cheap multi-modality;
- The claims are backed by proper experimental evidence.

### Significance

- The experiments over all the combinations of cross-modal tasks involving two of the fourmodalities considered are promising, and it’s impressive that these are obtained without the need for an expensive paired dataset covering all the involved modalities;
- Source code is provided for reproduction purposes.

**Weaknesses:**

###

- ULIP is already trained to be aligned with CLIP in the original paper. This work instead states it as a contribution;
    - while it may be the case that the framework improved this alignment, I find it should be stated clearly that this was indeed the case
- Clarity could be greatly improved, the amount and repetition of acronyms makes the overall paper hard to read and hardly a pleasing experience;
- After severeal reads, I still found it hard to decipher the “modality-centric consistency” part.
    - Being a core component of the framework, I find that the paper really needs to state it more clearly for the reader to understand;
- Even key messages in the introduction are somewhat vague and not immediately clear; e.g.
    - in “*from  the  training  data  perspective,  we  extract  various  modality-centric pseudo  data  pairs,  aiming  to  alleviate  the  semantic  bias  of  pseudo  pairs  in  Wang  et  al.  (2023b) and reflect MCR space more comprehensively.*” what does it mean for a data pair to be modality-centric? what semantic bias is the work referring to? what does it mean to reflect a MCR space comprehensively?
    - the whole period “*C-MCR employs data from overlapping modalities to aggregate semantic consistent embedding of non-overlapping modalities, thereby creating pseudo-pairs*” is not clear and should be rephrased for clarity.
    - These are only two examples, but this kind of ambiguous phrasing is very frequent in the text, unfortunately hindering my understanding of the contributions;
- The notation is not always clear, and sometimes the details are missing to fully comprehend the equations, such as the normalization factor of the softmax and the tilde symbols in (1) and following equations .
- Table 3, 4, 5 and 6 don’t report what metric is actually being shown.
- I can’t decipher what’s going on in Figure 1, especially in the left part. Overall the figure seems quite crowded, it probably would help splitting it in two figures. The caption doesn’t seem to help.

Overall, clarity is the principal reason I am inclined to reject as it prevents me from fully understanding the proposed method and contributions, therefore hindering my capability to assess its merits and drawbacks. Moreover, I find the work to tackle an interesting and impactful topic, and I would like it to be accessible to a broad audience; I find the current writing to pose a significant hurdle in this regard, and therefore hope it can be improved. I am more than willing to increase my score if this is properly addressed.

**Questions:**

### Questions
- What’s the denominator in the softmax computation in (1)? is it over all the samples in the batch?
- I don’t understand what the tilde means in Eq (1) and the following ones, how does $\tilde{t}$ differ from $t$?
- in 3.2.1., it mentions three kinds of semantically consistent embeddings (audio-centric, text-centric, image-centric) (here the e.g. should actually be a i.e.) but then four objects are given, $\{\tilde{a}_i^A, \tilde{t}_i^A, \tilde{t}_i^I, \tilde{v}_i^I\}$. What’s the explanation here?
### Typos
- 3.2.1. header: centirc —> centric
- The text sometimes refers to base MCR as based MCR

---

> ### Author Response · Authors · 2023-11-20
> **Response to Reviewer pnPG (1/2)**
>
> Thank you very much for your appreciation of our work and your valuable suggestions on the presentation of our manuscript. In our newly submitted PDF, we have carefully revised the presentation of most paragraphs to improve its clarity and readability. We hope the new submission can solve your concerns about writing.
>
> ### **W1: statements about ULIP**
>
> ULIP aligns a 3D encoder to an image-text model called SLIP [1] (not CLIP) through 3D-image-text data. Ex-MCR only uses the aligned 3D-image representation of ULIP and extends it to a different image-text model (i.e., CLIP) via the paired-data-free way. So we are not reproducing or refining the 3D-text alignment of ULIP, but building a new alignment between the 3D representation and new image-text pre-training.
>
> [1] Slip: Self-supervision meets language image pre-training. Norman Mu, Alexander Kirillov, David Wagner, and Saining Xie. ECCV 2022
>
> ### **W2: repetition of acronyms**
>
> Thanks for your suggestions, in the newly submitted paper we carefully refine the usage of acronyms to improve readability. The modifications of acronyms can be summarized as:
>
> 1. Avoiding confusion due to many similar acronyms. Replace “MCR” with "space", "intra-MCR alignment" with "intra-space alignment", and "inter-MCR alignment" with"inter-space alignment". Remove unnecessary acronyms, like “MLP”, “InfoNCE Loss” and “L2 Loss”
> 2. Use "audio-text space" "image-text space" and "image-3D space" in the introduction instead of using "CLAP", "CLIP", and "ULIP" to directly explain the aligned modalities in each space.
>
> ### **W3 & W4: Statements in the “modality-centric consistency” part**
>
> We reorganize and rewrite the entire "modality-centric consistency" part in the new submission.
>
> In the introduction, we avoid directly giving many self-defined terms without explicit explanation, such as: “modality-centric”, “semantic bias” and “reflect an MCR space comprehensively”. The so-called modality $\mathcal{A}$-centric pseudo data means using only data of modality $\mathcal{A}$ to retrieve semantically similar data in other modalities, while "semantic bias" means that one modality cannot be fully represented by another modality (giving examples of audio and images to support), and "various modality-centric data" denotes combining the pseudo pairs retrieved by different modalities.
>
> In the first and second paragraphs of the Section 3.2.1, we define the "single modality-centric data" and "various modality-centric data", and provide examples of image of “mushroom” and audio of “wind noise” to illustrate that single modality-centric data cannot comprehensively reflect representations of different modalities in different MCR spaces.
>
> ### **W5 & Q1 & Q2: unclear notation**
>
> The $softmax(\cdot)$ in Equation 1 and Equation 2 is over all the samples in used datasets.
>
> The InfoNCE function in Equation 6 is calculated on all the samples in a training batch.
>
> The tilde symbols mean the features are processed to be semantically consistent.
>
> ### **W6: The metrics in Table 3,4,5,6**
>
> At the caption of Table 3 and the beginning of Section 4.5, we add descriptions about the metrics used in Tables 3, 4, 5, and 6. In ablation experiments, we report the mAP metrics on audio-image retrieval (AVE) and audio-text retrieval (AudioCaps). In the Appendix, we provide detailed ablation experiment results on all the datasets and metrics.
>
> ### **W7: Too crowded Figure 1**
>
> We redesign Figure 1 and rewrite its caption. Figure 1 is divided into two sub-figures (a) (b).
>
> **To improve the simplicity of the figure, we made the following changes to Figure 1 (a):**
>
> 1). Remove unnecessary text such as "pull close" and "pull close & push away" and replace them with a gray bold dash line.
>
> 2). Replace text such as "audio features" "CLAP Text features" with corresponding vector symbols. On the one hand, keeping there are only vector symbols, modules, and lines in the pipeline figure, can display the calculation process more clearly. On the other hand, it echoes the notation definitions in Section 3.2 and helps the reader to quickly understand the meaning of the defined notations.
>
> 3). Frame the base-MCR and two leaf-MCRs into three boxes, which not only improves the aesthetics and simplicity, but also emphasizes that the three pre-trained spaces are separate.
>
> 4). Add explanations about some symbols at the bottom to help understand the illustrations.
>
> **For Figure 1 (b), the subfigure about various modality-centric data**
>
> we cancel the repeated "text-centric consistency", "audio-centric consistency", and "image-centric consistency" which are difficult to understand. More emphasis is placed on combining pseudo-data pairs from different sources to form a data pool.
>
> For captions, we do not broadly explain the entire picture, but provide relevant explanations for each sub-picture. And use more space to explain the more complex Figure 1 (b).

---

> > ### Comment · Reviewer_pnPG · 2023-11-23
> >
> > I thank the authors for their rebuttal. While I find the overall clarity and writing of the paper to still be improvable, I find the modifications to already point in the right direction and be enough for the reader to understand the method. I am confident the paper will be further improved for the camera-ready version. I am therefore raising my score to 6.

---

> ### Author Response · Authors · 2023-11-20
> **Response to Reviewer pnPG (2/2)**
>
> ### **Q3: The explanation of $\{\tilde{\mathbf{a}}_i^A, \tilde{\mathbf{t}}_i^A, \tilde{\mathbf{t}}_i^I, \tilde{\mathbf{v} }_i^I\}$**
>
> Since the pseudo data pairs retrieved using different modalities are shuffled to form the final data pool, we do not distinguish symbolically between data pairs from different sources. The $\{ \tilde{\mathbf{a}}_i^A, \tilde{\mathbf{t}}_i^A, \tilde{\mathbf{t}}_i^I, \tilde{\mathbf{v} }_i^I \}$ represents a pseudo-pair sample in the final data pool, and this sample can be generated from any source.

---

> ### Author Response · Authors · 2023-11-22
> **Deadline coming. Looking forward to your feedback.**
>
> Dear Reviewer pnPG,
>
> Thanks again for your comments. We would like to kindly remind you that we tried our best to address the concerns you raised.
>
> We kindly request your feedback as the rebuttal deadline is approaching in less than 1 day. We would be happy to discuss in detail if you have additional comments about our paper.
>
> Best regards, Authors

---

### Author Response · Authors · 2023-11-21
**Looking forward to further feedback**

Dear Reviewers,

We hope that our response and new submission have effectively addressed your comments and concerns. As we approach the deadline for the discussion period, we wanted to kindly ask you if you could review our response. We would greatly appreciate it if you could let us know of further questions.

Paper 3649 authors